# Size-Noise Tradeoffs in Generative Networks

**Bolton Bailey**     **Matus Telgarsky**
`{boltonb2,mjt}@illinois.edu`
University of Illinois, Urbana-Champaign

## Abstract

This paper investigates the ability of generative networks to convert their input noise distributions into other distributions. Firstly, we demonstrate a construction that allows ReLU networks to increase the dimensionality of their noise distribution by implementing a "space-filling" function based on iterated tent maps. We show this construction is optimal by analyzing the number of affine pieces in functions computed by multivariate ReLU networks. Secondly, we provide efficient ways (using $\mathrm{polylog}(1/\epsilon)$ nodes) for networks to pass between univariate uniform and normal distributions, using a Taylor series approximation and a binary search gadget for computing function inverses. Lastly, we indicate how high dimensional distributions can be efficiently transformed into low dimensional distributions.

## 1  Introduction

This paper focuses on the representational capabilities of generative networks. A generative network models a complex target distribution by taking samples from some efficiently-sampleable noise distribution and mapping them to the target distribution using a neural network. What distributions can a generative net approximate, and how well? Larger neural networks or networks with more noise given as input have greater power to model distributions, but it is unclear how the use of one resource can make up for the lack of the other. We seek to describe the relationship between these resources.

In our analysis, we make a few assumptions on the structure of the network and the noise. We focus on the two most standard choices for noise distributions: The normal distribution, and the uniform distribution on the unit hypercube [Arjovsky et al., 2017]. Henceforth, we will use the term "uniform distribution" to refer to the uniform distribution on unit hypercubes, unless otherwise specified. We look specifically at the case where the generative network is a fully-connected network with the ReLU activation function (without weight sharing). The notion of approximation we use is the *Wasserstein distance*, introduced for generative networks in Arjovsky et al. [2017], which is defined as follows:

**Definition 1.** *For two distributions $\mu$ and $\nu$ on $\mathbb{R}^d$, their Wasserstein distance is defined as*

$$W(\mu,\nu) := \inf_{\pi \in \Pi(\mu,\nu)} \int |x - y| d\pi(x,y),$$

*where $\Pi(\mu,\nu)$ is the set of joint distributions having $\mu$ and $\nu$ as marginals.*

Our results fall into three regimes, each covered in its own section:

**Section 2: The case where the input dimension is less than the output dimension.**
  In this regime, we prove tight upper and lower bounds for the task of approximating higher dimensional uniform distributions with lower dimensional distributions in terms of the average width $W$ and depth (number of layers) $L$ of the network. The bounds are tight in the sense that both give an accuracy of the form $\epsilon = O(W)^{-O(L)}$ (keeping input and output dimensions fixed). Thus, this gives a good idea of the asymptotic behavior in this regime: Error exponentially decays with the number of layers, and polynomially decays with the number of nodes in the network.

**Section 3: The case where the input and output dimensions are equal.**
> In this regime, we give constructions for networks that can translate between single dimensional uniform and normal distributions. These constructions incur $\epsilon$ error in Wasserstein distance using only $\mathrm{polylog}(1/\epsilon)$ nodes.

**Section 4: The case where the input dimension is greater than the output dimension.**
> In this regime, we show that even with trivial networks, increased input dimension can sometimes improve accuracy.

In the course of proving the above results, we show several lemmas of independent interest.

**Multivariable affine complexity lemma.**
> For a function $f : \mathbb{R}^{n_0} \to \mathbb{R}^d$ computed by a neural network with $N$ nodes and $L$ layers and ReLU nonlinearities, the domain of $f$ can be partitioned into $O\left(\frac{N}{n_0 L}\right)^{n_0 L}$ convex (polyhedral) pieces such that $f$ is affine on each piece. This is extends prior work, which considered networks with only univariate input [Telgarsky, 2016].

**Taylor series approximation.**
> Univariate functions with quickly decaying Taylor series, such as $\exp, \cos$, and the CDF of the standard normal, can be approximated on domains of length $M$ with networks of size $\mathrm{poly}(M, \ln(1/\epsilon))$. This idea was been explored before by Yarotsky [2017]; the key difference between this work and the prior is that our results apply directly to arbitrary domains.

**Function inversion through binary search.**
> The inverses of increasing functions with large enough slope can be approximated efficiently, provided that the functions themselves can be approximated efficiently. While this technique does not provide uniform bounds on the error, we show that it provides approximations that are good enough for generative networks to have low error.

Detailed proofs of most theorems and lemmas can be found in the appendix.

## 1.1 Related Work

Generative networks have become popular in the form of Generative Adversarial Networks (GANs), introduced by Goodfellow et al. [2014]; see for instance [Creswell et al., 2018] for a survey of various GAN architectures. GANs are trained using a discriminator network, an auxiliary neural network which tries to prove the distance from the simulated distribution to the true data distribution is large. The generator is trained by gradient descent to minimize the distance given by the adversary network. Wasserstein GANs (or WGANs) are GANs which use an approximation of the Wasserstein distance as this notion of distance. The concept of Wasserstein distance comes out of the theory of optimal transport, as discussed in Villani [2003], and its use as a performance metric is expounded in Arjovsky et al. [2017]. WGANs have shown success in various generation tasks [Osokin et al., 2017, Donahue et al., 2018, Chen and Tong, 2017]. While this paper uses the Wasserstein distance as a performance metric, we are not concerned with the training process, only the representational capabilities of the networks.

Many of the results in this paper build out of the results on the representational power of neural nets as function approximators. These results first focused upon approximating continuous functions with a single hidden layer [Hornik et al., 1989, Cybenko, 1989], but recently branched out to deeper networks [Telgarsky, 2016, Eldan and Shamir, 2016, Yarotsky, 2017, Montufar et al., 2014]. A concurrent work in this area is Zhang et al. [2018], which uses tropical geometry to analyze deep networks. This work produced a result on the number of affine pieces of deep networks [Zhang et al., 2018, Theorem 6.3], which matches our bound in Lemma 1. This bound was originally suggested in Montufar et al. [2014]. The present work relies upon some of these recent works (e.g., affine piece counting bounds, approximation via Taylor series), but develops nontrivial extensions (e.g., multivariate inputs and outputs with tight dimension dependence, less benign Taylor series).

The representational capabilities of generative networks have previously been studied by Lee et al. [2017]. That paper provides a result for the representation capabilities of deep neural networks in terms of "Barron functions", first described in Barron [1993], which are functions with certain constraints on their Fourier transform. Lee et al. [2017] showed that compositions of these Barron

functions could be approximated well by deep neural networks. Their main result with respect to the representation of distributions was that the result of mapping a noise distribution through a Barron function composition could be approximated in Wasserstein distance by mapping the same noise distribution through the neural network approximation to the Barron function composition. These techniques do not readily permit the analysis of target distributions which are not images of the input space under these Barron functions.

The Box-Muller transform [Box et al., 1958] is a computational method for simulating bivariate normal distributions using uniform distributions on the unit (2-dimensional) square. The method is a general algorithm, but it is possible to simulate the transform with specially constructed neural nets, to prove theorems similar to those in section 3. In fact this was our original approach; an overview of the Box-Muller implementation can be found in section 3.

## 1.2 Notation for Neural Networks

We define a neural network with $L$ layers and $n_i$ nodes in the $i$th layer as a composition of functions of the form

$$A_L \circ \sigma_{n_{L-1}} \circ A_{L-1} \circ \sigma_{n_{L-2}} \circ \cdots \circ \sigma_{n_1} \circ A_1.$$

Here $A_i : \mathbb{R}^{n_{i-1}} \to \mathbb{R}^{n_i}$ is an affine function. That is, $A_i$ is the sum of a linear function and a constant vector. $\sigma_k : \mathbb{R}^k \to \mathbb{R}^k$ is the $k$-component pointwise ReLU function, where the ReLU is the map $x \mapsto \max\{x, 0\}$. The total number of nodes $N$ in a network is the sum $\sum_{i=0}^{L} n_i$. We will sometimes use $n = n_0$ to refer to the input dimension and $d = n_L$ to refer to the output dimension.

Since a neural network is a composition of piecewise affine functions, it is piecewise affine. The number of affine pieces of a function $f$ will be denoted $N_A(f)$ or just $N_A$.

When $\mu$ is a distribution, we will adopt the notation of Villani [2003] and use $f\#\mu$ to denote the pushforward of $\mu$ under $f$, i.e., the distribution obtained by applying $f$ to samples from $\mu$. We will use $U(A)$ to denote the uniform distribution on a set $A \subset \mathbb{R}^n$, and $m(A)$ to denote the Lebesgue measure of that set. We will use $\mathcal{N}$ to denote a normal distribution, which will always be centered on the origin and have unit covariance matrix.

## 2 Increasing the Dimensionality of Noise

How easy is it to create a generator network that can output more dimensions of noise than it receives? It is common in practice to use a far greater output dimension. Here, we give both upper and lower bounds showing that an increase in dimension can require a large, complicated network.

### 2.1 Constructions for the Uniform Hypercube

For this section, we restrict ourselves to the case of input and output distributions which are uniform. To motivate our techniques, we can simplify our problem even further: We could ask how one might approximate a uniform distribution on the unit square using the uniform distribution on the unit interval. We see that we are limited by the fact that the range of the generator net is some one-dimensional curve in $\mathbb{R}^2$, and so the distribution that the generator net produces will have to be supported on this curve. We will want each point of the unit square to be close to some point on the curve so that the mass of the square can be transported to the generated distribution. We are therefore led to consider some kind of (almost) space filling curve. An excellent candidate is the graph of the iterated tent map, shown in Figure 2.1. This function has been useful in the past [Montufar et al., 2014, Telgarsky, 2016] since it is highly nonlinear and it can be shown that neural networks must be large to approximate it well. We can create a construction for the univariate to multivariate network which uses tent maps of varying frequencies to fill space.

The tentmap construction, which appears in [Montufar et al., 2014] and is given in full in the appendix, achieves the following error:

**Theorem 1.** *Let $\mu$ and $\nu$ respectively denote uniform distributions on $[0, 1]$ and $[0, 1]^d$. Given any number of nodes $N$ and number of layers $L$ satisfying $N > dL$, we can construct a generative*

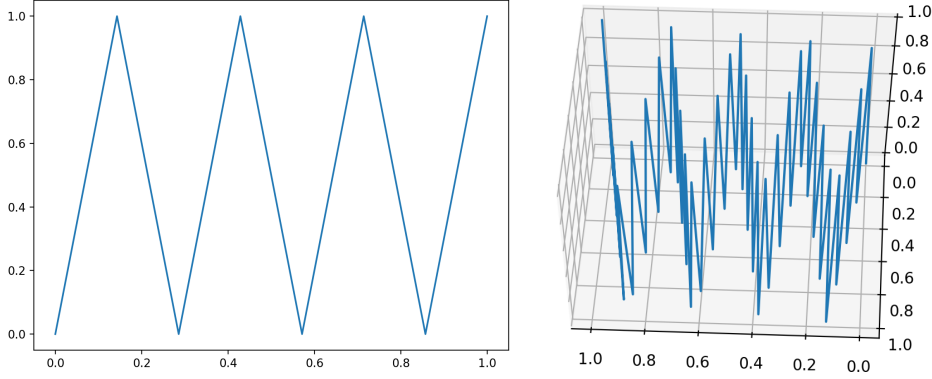

Figure 1: Examples of paths that come near every point in the unit square and the unit cube.

network $f : [0,1] \to [0,1]^d$ such that

$$W(f\#\mu, \nu) \leq \sqrt{d} \left\lfloor \frac{N - dL}{L} \right\rfloor^{-\left\lfloor \frac{L}{d-1} \right\rfloor}. \tag{1}$$

Thus, as the size of the network grows, the base in Equation 1 grows proportionally to the average width of the network, and the exponent grows proportionally to the depth of the network (while being inversely proportional to the number of outputs). The $N > dL$ requirement comes from using some nodes to carry values forward — If we were to allow connections between non-adjacent layers, this requirement would go away and $N$ would replace $N - dL$ in the theorem statement.

We now consider the case where our input noise dimension is larger than 1. In this case, we see that one possible construction involves dividing the output dimensions evenly amongst the input dimensions and then placing multiple copies of the above described construction in parallel. This produces the following bound:

**Theorem 2.** *Let $\mu$ and $\nu$ respectively denote uniform distributions on $[0,1]^n$ and $[0,1]^d$. Given any number of nodes $N$ and number of layers $L$ satisfying $N > dL$, we can construct a generative network $f : [0,1]^n \to [0,1]^d$ such that*

$$W(f\#\mu, \nu) \leq \sqrt{d} \left\lfloor \frac{N - dL}{nL} \right\rfloor^{-\left\lfloor \frac{L}{\lceil \frac{d-n}{n} \rceil} \right\rfloor} = O\left(\frac{N}{nL}\right)^{-O\left(\frac{nL}{d}\right)},$$

*where the big-O hides factors of $d$ in the base, and constant factors in the exponent.*

Note that this generalizes Theorem 1. The proof can be found in the appendix. This bound is at its tightest when $d$ is a multiple of $n$, in which case $\frac{d-n}{n}$ is an integer, and the exponent matches exactly that in the lower bound determined later. The construction works more smoothly with this even divisibility because the output nodes can be evenly split among the inputs, and it is easier to parallelize the construction.

## 2.2 Lower Bounds for the Uniform Box

We now provide matching lower bounds. For this, it suffices to count the affine pieces. Bounds on the number of affine pieces have been proved before, but only with univariate input [Telgarsky, 2016]; here we allow the network input to be multidimensional.

**Lemma 1.** *Let $f : \mathbb{R}^{n_0} \to \mathbb{R}^d$ be a function computed by a neural network with at most $N$ total nodes and $L$ layers. Then the domain of $f$ can be divided into $N_A$ convex pieces on which the function is affine, where*

$$N_A \leq \left(e \frac{N}{n_0 L} + e\right)^{n_0 L}.$$

This lemma has also been proven in concurrent work [Zhang et al., 2018] using the techniques of tropical geometry. Our proof works essentially by induction on the number of layers: We look at the set of possible activations of the $i$th layer, we see that it is a union of convex affine sets of dimension at most $n_0$. The application of the ReLU maps each of these convex affine sets into $O(n_i)^{n_0}$ convex affine sets, where $n_i$ is the number of nodes in the $i$th layer.

The one-dimensional tent map construction tells us that for a given number of nodes and number of layers, we can construct a function with a number of affine pieces bounded by the size of the network. When constructing multidimensional input networks with a high number of affine pieces, we can always parallelize several of these tentmaps to get a map with the product of the number of pieces in the individual networks. What this lemma guarantees is that, up to a constant factor in the number of nodes, this construction is optimal for producing as many affine pieces as possible. This gives us confidence that our parallelized tent map construction for low-dimensional uniform to high-dimensional uniform may be close to optimal.

To show that our construction is optimal, we need to show that it approximates the high-dimensional uniform distribution about as accurately as any piecewise affine function with the same number of pieces $N_A$. To do this, we will use the fact that the range of a piecewise affine function is a subset of the union of ranges of its constituent affine functions. We then show that any distribution on a union like this is necessarily distant from the target uniform distribution.

**Theorem 3.** *Let $B$ be a bounded measurable subset of $\mathbb{R}^d$ of radius $l$, let $f : \mathbb{R}^n \to \mathbb{R}^d$ be piecewise affine with $n < d$, and let $P$ be any distribution on $\mathbb{R}^n$. The Wasserstein distance between $f\#P$ and the uniform distribution $U_B$ on $B$ has the following lower bound:*

$$W(U_B, f\#P) \geq k \left( l^{-n} \frac{m(B)}{N_A} \right)^{\frac{1}{d-n}},$$

*where $k$ depends on $n$ and $d$.*

Note that our technique can produce bounds not just for the unit cube on $n$-dimensions but for any uniform distribution on any bounded subset of $\mathbb{R}^n$ such as the sphere. When we combine this with our analysis of $N_A$ in Lemma 1, we get a lower bound result for a given number of nodes and layers.

**Theorem 4.** *Let $\mu$ and $\nu$ respectively denote uniform distributions on $[0,1]^n$ and $[0,1]^d$. Given any number of nodes $N$ and number of layers $L$, for any generative network $f : [0,1]^n \to [0,1]^d$, we have*

$$W(f\#\mu, \nu) \geq k \left( e\frac{N}{nL} + e \right)^{-\frac{nL}{d-n}} = O\left( \frac{N}{nL} \right)^{-\frac{nL}{d-n}},$$

*where the big-O hides factors of $n$ and $d$ in the base.*

*Proof.* This follows from applying Theorem 3 taking $f$ as a neural network with the affine piece bound from Lemma 1, and $P$ as $\mu$, the uniform distribution on $[0,1]^n$. $\square$

## 3 Transporting between Univariate Distributions

The two most common distributions used in generative networks in practice are the uniform distribution, and the normal. How easily can one of these distributions can be used to approximate the other? We will deal with the simplest case where our input and output distributions are one-dimensional. If we can construct a neural net for this case, we can parallelize multiple copies of the net if we want to move between normal and uniform distributions in higher dimensions.

### 3.1 Approximation of a Uniform Distribution by a Normal

Perhaps the simplest idea for approximating a uniform distribution with a generative network with normal noise is to let the network approximate $\Phi$, the cumulative distribution function of the normal. To approximate $\Phi$, we will approximate its Maclaurin series:

$$\Phi(z) = \frac{1}{2} + \frac{1}{\sqrt{2\pi}} \sum_{n=0}^{\infty} \frac{(-1)^n z^{2n+1}}{n!(2n+1)2^n}.$$

This series has convergence properties which allow a network based on its truncation to work.

Yarotsky [2017, Proposition 3c] showed that $f : (x, y) \mapsto xy$ over $[-M, M]^2$ can be efficiently approximated by neural networks, in the sense that there is a network with $O(\ln(1/\epsilon) + \ln(M))$ nodes and layers computing a function $\hat{f}$ with $|f - \hat{f}| \leq \epsilon$. Yarotsky [2017] uses this to show that certain functions with small derivatives could be approximated well. We will show a similar result, which depends on the good behavior of the Taylor series of $\Phi$.

Naturally, if $\tilde{f}$ is a neural network approximating $f$ and $\tilde{g}$ approximates $g$, then we can compose these approximations to get an approximation of the composition. In particular, if $g$ has a Lipschitz constant, then the composition approximation will depend on this Lipschitz constant and the accuracies of the individual approximations. A "composition lemma" to this effect is included in the appendix as Lemma 8. We will use this idea several times to construct a variety of function approximations.

We will now consider the method of approximating functions by approximating their Taylor series with neural networks. To do this, we first demonstrate a network which takes a univariate input $x$ in $[-M, M]$ and returns the multivariate output $(x^0, x^1, x^2, \ldots, x^n)$.

**Theorem 5.** *The function $f : x \mapsto (x^0, \ldots, x^n)$ on $[-M, M]$ can be computed uniformly to within $\epsilon$ by a neural network of size $\mathrm{poly}(n, \ln(M), \ln(1/\epsilon))$.*

The proof relies on iteratively composing the multiplication function $x^i = x^{i-1} \cdot x$ using the "composition lemma" to get each of the $x^i$.

Now that we know the size required to approximate the powers of $x$, we may use this to approximate the Maclaurin series of $\Phi$.

**Theorem 6.** *The function $\Phi$ can be approximated uniformly to within $\epsilon$ by a network of size $\mathrm{poly}(\ln(1/\epsilon))$.*

To show this we apply Theorem 5 with a suitable choice of $M$ and $n$ and then use the monomial approximations to get a Taylor approximation of $\Phi$. Knowing that we can approximate $\Phi$ well, we can give a precise bound on the Wasserstein distance of this construction.

**Theorem 7.** *We can construct a generative network with $\mathrm{polylog}(1/\epsilon)$ nodes and univariate normal noise that can output a distribution with Wasserstein distance $\epsilon$ from uniform.*

*Proof.* Using Theorem 6, let $\tilde{\Phi} : \mathbb{R} \to [0, 1]$ approximate $\Phi$ uniformly to within $\frac{\epsilon}{2}$. Consider the coupling $\pi$ between the output of this network and the true uniform distribution which consists of pairs $(\Phi(z), \tilde{\Phi}(z))$, where $z$ is normally distributed:

$$W\left(U([0, 1]), \tilde{\Phi}\#\mathcal{N}\right) = W\left(\Phi\#\mathcal{N}, \tilde{\Phi}\#\mathcal{N}\right) \leq \int_{\mathbb{R}} |\Phi(z) - \tilde{\Phi}(z)| \cdot \frac{1}{\sqrt{2\pi}} e^{-z^2/2} dz.$$

But since $|\Phi - \tilde{\Phi}|$ is less than $\epsilon$ everywhere, this integral is no more than $\epsilon$, so we can indeed create a generative network of $\mathrm{polylog}(1/\epsilon)$ nodes for this task. $\square$

## 3.2 Approximation of a Normal Distribution by Uniform

Having shown that normal distributions can approximate uniform distributions with $\mathrm{polylog}(1/\epsilon)$ nodes, let's see if the reverse is true. For this we'll need a few lemmas.

For analytic convenience, a few of our intermediate constructions will use networks with both ReLU activations, as well as step functions $H(x) = \mathbb{1}[x > 0]$. Networks with these two allowed activations have a convenient property which allows them to be used to study vanilla ReLU networks: If there is a ReLU/Step network approximating a function $f$ uniformly, then $f$ can be uniformly approximated by a comparably-sized network on all but an arbitrarily small positive subset of its domain.

**Lemma 2.** *Let $\mu$ be a measure, and $A$ a measurable set with $\mu(A) < \infty$. Suppose $f : \mathbb{R}^n \to \mathbb{R}^d$ can be approximated uniformly to within $\epsilon$ on $A$ by a function $\tilde{f}$ computed by a ReLU/Step network with $N$ nodes. Then for any $\zeta > 0$, there exists a ReLU network with $O(N)$ nodes which approximates $f$ to within $2\epsilon$ on a set $B$ where $A \setminus B$ has measure less than $\zeta$.*

*Proof.* Note that while a ReLU neural network cannot implement the step function, it can implement the following approximation:

$$s_\delta(x) = \begin{cases} 0 & \text{if } x \leq 0, \\ x/\delta & \text{if } 0 \leq x \leq \delta, \\ 1 & \text{if } x > \delta. \end{cases}$$

In fact, this approximation to the step function can be implemented with a 4-node ReLU network. If we replace every step function activation node in our architecture with a copy of this four node network, we get an architecture of size $O(N)$. With this architecture, we can compute each of a sequence $(f_n)$ of functions, where in $f_n$, all step functions from our old network are replaced by $s_{1/n}$. For any $x$ in $A$, consider the minimum positive input to the step function which occurs in the computation graph. If $\delta = 1/n$ is less than this minimum, then $f_n(x) = f(x)$, so this sequence converges pointwise to $\tilde{f}$. Egorov's theorem [Kolmogorov and Fomin, 1975, pp. 290, Theorem 12] now tells us that $(f_n)$ converges to $\tilde{f}$ uniformly on a set $B$ that satisfies the $\mu(A \setminus B) < \zeta$ requirement. Thus, there is an $f_n$ that approximates $\tilde{f}$ to within $\epsilon$ uniformly on this $B$, and $f_n$ therefore approximates $f$ uniformly on $B$ to within $2\epsilon$. □

This lemma has a useful application to generative networks: If we make $\zeta$ sufficiently small, the mass of the noise distribution on $A \setminus B$ is arbitrarily small. Therefore, we can make $\zeta$ small enough that the impact of the mistake region on the Wasserstein distance is negligible.

We now would like to approximate some function that maps the uniform distribution to the normal in this powerful format. Complementing the use of the normal CDF $\Phi$ in the previous subsection, here we will use its inverse $\Phi^{-1}$. Since we conveniently have already proved that $\Phi$ is efficiently approximable, we would like a general lemma that allows us to invert this.

**Lemma 3.** *Let $f : [a, b] \to [c, d]$ be a strictly increasing differentiable function with $f'$ greater than a constant $L$ everywhere, and let $f$ be approximated to within $\epsilon$ by a network of size $N$. Then (for any $\zeta > 0$), $f^{-1}$ can be approximated to within $(b - a)2^{-t} + \epsilon L$ on (all but a measure $\zeta$ subset of) $[c, d]$ by a network of size $O(tN)$.*

The proof of this lemma constructs a neural network that executes $t$ iterations of a binary search on $[a, b]$, using $t$ copies of the approximation to $f$ to decide which subinterval to narrow in on. Applying this lemma to our approximation theorem for the normal CDF gives us an approximation of the inverse of the normal CDF.

**Theorem 8.** *For any $\zeta > 0$, the function $\Phi^{-1}$ can be approximated to within $\epsilon$ by a network of size $\text{polylog}(1/\epsilon)$ on $[\Phi(-\ln(1/\epsilon)), \Phi(\ln(1/\epsilon))] \setminus A$ where $A$ is of measure $\zeta$.*

*Proof.* By Theorem 6 we can get the normal CDF $\Phi$ to within $\epsilon^{\ln(1/\epsilon)+1}$ with $\text{polylog}(1/\epsilon)$ nodes. Using Lemma 3 with $t = O(\ln(1/\epsilon))$, if we choose $a = -\ln(1/\epsilon), b = \ln(1/\epsilon)$ then the Lipschitz constant of $\Phi^{-1}$ on this interval is

$$\Phi'(\ln(1/\epsilon))^{-1} = O(e^{\ln(1/\epsilon)^2/2}) = O(\epsilon^{-\ln(1/\epsilon)}),$$

and so Lemma 3 gives a total error on the order of $\epsilon$. □

With this approximation, we can get a generative network approximation. Since the tails of the normal distribution are small, we can ignore them by collapsing the mass of the tails into a bounded interval. Then, by setting $\zeta$ sufficiently small that the Wasserstein distance contributed by the error region is negligible, our approximation can be shown to be within $\epsilon$ of the normal.

As a final lemma, we note the following observation

**Proposition 1.** *For two distributions on $\mathbb{R}$, their Wasserstein distance is equal to the $L^1$ integral of the difference of their CDFs.*

For a proof, see [Villani, 2003, remark 2.19.ii]. For an intuition, note that moving a mass $m$ from $a$ to $b$ on a one-dimensional distribution changes the CDF of the distribution on $[a, b]$ by $m$.

With these in place, we use get a bound for the uniform to normal construction.

**Theorem 9.** *A generative network with* $\mathrm{polylog}(1/\epsilon)$ *nodes and univariate uniform noise can output a distribution with Wasserstein distance $\epsilon$ from a normal distribution.*

The proof is an application of Theorem 8 and Proposition 1.

### 3.2.1 The Box-Muller Transform

We've established the bound we sought (approximation of a normal distribution via uniform), but in this section we'll show that a curious classical construction also fits the bill, albeit in two dimensions. The *Box-Muller transform* [Box et al., 1958] comes from the observation that if $X_1$ and $X_2$ are two independent uniform distributions on the unit interval, then if we define

$$Z_1 := \sqrt{-2\ln(X_1)}\cos(2\pi X_2) \qquad \text{and} \qquad Z_2 := \sqrt{-2\ln(X_1)}\sin(2\pi X_2), \qquad (2)$$

then $Z_1, Z_2$ are independent and normally distributed. Equation 2 comes from the interpretation of $\sqrt{-2\ln(X_1)}$ and $2\pi X_2$ as $r$ and $\theta$ in a polar-coordinate representation of the pair of normals. While this method is not as powerful as the CDF approximation method, in that it requires two dimensions of uniform noise in order to work, it still suggests an idea for a similar theorem to Theorem 8.

**Theorem 10.** *For any $\zeta > 0$, the function $X_1, X_2 \mapsto \sqrt{-2\ln(X_1)}\cos(2\pi X_2), \sqrt{-2\ln(X_1)}\sin(2\pi X_2)$ can be approximated to within $\epsilon$ by a network of size* $\mathrm{polylog}(1/\epsilon)$ *on $[0,1]^2 \setminus A$ where $A$ is of measure $\zeta$.*

We provide the following proof sketch:

- The $\cos$ and $\sin$ functions (and the $\exp$ function) can be efficiently computed for much the same reason that $\Phi$ can: Their Taylor expansion coefficients decay rapidly.
- The $\ln$ function can be approximated in $[1/2, 3/2]$ using the Taylor series for $\ln(1+x)$. For inputs outside this interval, we can repeatedly multiply double/halve the input until we reach $[1/2, 3/2]$, use the approximation we have, then add in a constant depending on the number of times we doubled or halved.
- The square root function, and in fact all functions of the form $x \mapsto x^\alpha$ for $\alpha > 0$, can be approximated using the approximations for $\exp$ and $\ln$ and the identity $x^\alpha = \exp(\alpha \ln(x))$.
- Putting these together, as well as the approximation for products from Yarotsky [2017], we get the result.

## 4 From Many Dimensions to One

This section will complete the story by seeing what is gained in transporting many dimensions into one.

To begin, let's first reflect on the bounds we have. So far, we have shown upper bounds on neural network sizes that are polylogarithmic in $1/\epsilon$. A careful analysis of the previous subsection shows that the construction uses $O(\ln^5(1/\epsilon))$ for normal to uniform and $O(\ln^{18}(1/\epsilon))$ for the uniform to normal. We would like to know how close to optimal these exponents are. The goal of this subsection is to quickly establish that the lower bound for this exponent is at least 1. To do this, we will make some use of the affine piece analysis from Section 2.

Note that piecewise affine functions acting on the uniform distribution have structure in their CDF, since they are a mixture of distributions induced by each individual affine piece:

**Proposition 2.** *For a piecewise affine function $f : [0,1] \to \mathbb{R}$ with $N_A$ pieces, the CDF of the a distribution $f\#U([0,1])$ is a piecewise affine function with at most $N_A + 2$ pieces.*

So if we can establish a bound on the accuracy with which a piecewise affine function can approximate the normal CDF, we can use the univariate affine pieces lemma above to lower bound the accuracy of any uniform univariate noise approximation of the normal. A helpful bound is given in Safran and Shamir [2016], from which we get:

**Lemma 4.** *Let $f$ be a univariate piecewise affine function with $N_A$ pieces. Then*

$$\int_a^b |\Phi(x) - f(x)|dx \geq \frac{K}{N_A^4}$$

*for some constant $K$.*

Putting this together with Proposition 2 and Lemma 1, we see:

**Theorem 11.** *A generative network taking uniform noise can approximate a normal with Wasserstein accuracy exponential in the number of nodes.*

Or in other words, approximation to accuracy $\epsilon$ requires at least $O(\log(1/\epsilon))$ nodes.

Clearly, if we wish to approximate a low-dimensional uniform distribution with a higher-dimensional one, all we need to do is ignore some of the inputs and spit the others back out unchewed. The same goes for normal distributions. Is there any benefit at all to additional dimensions on input noise when the target distribution is a lower dimension?

Interestingly, the answer is yes. Considering the case of approximating a univariate normal distribution with a high dimensional distribution, we note that there is the simplistic approach which involves summing the inputs and reasoning that the output is close to a normal distribution by the Berry-Esseen theorem.

**Theorem 12.** *The distribution given by summing $n$ uniform random variables on $[0,1]$ and normalizing the result has a Wasserstein distance of $O(\frac{1}{\sqrt{n}})$ from the standard normal distribution.*

Note that the above approach does not use any nonlinearity at all. It simply takes advantage of the fact that projecting a hypercube onto a line results in an approximately normal distribution. This theorem suggests another way of approaching Theorem 9: Use the results of section 2 to increase a 1-dimensional uniform distribution to a $d$-dimensional uniform distribution, then apply Theorem 12 as the final layer of that construction to get an approximately normal distribution. Unfortunately, this technique does not prove the $\text{polylog}(1/\epsilon)$ size: it is necessary for $\frac{1}{\sqrt{d}} \approx \epsilon$, which means the size of the network (indeed, even the size of the final layer of the network) is polynomial in $1/\epsilon$.

## 5  Conclusions and Future Work

One might ask with regards to Section 3 if there are more efficient constructions than the ones found in this section, since there is a gap between the upper and lower bounds. There are other approaches to the uniform to normal transformation, such as the Box-Muller method [Box et al., 1958] we discuss. Future work could modify this or other methods to tighten the bounds found in this section.

An interesting open question is whether the results of Section 3 can be applied more generally to multidimensional distributions. Suppose for example that we have a neural network that pushes a univariate uniform distribution into a univariate normal distribution. We can take $d$ copies of this network in parallel to get a network which takes $d$-dimensional uniform noise, and outputs $d$-dimensional normal noise. Is a parallel construction of the form described here the *most efficient* way to create a network that pushes forward a $d$-dimensional uniform distribution to a $d$-dimensional normal? For that matter, if $f : \mathbb{R}^d \to \mathbb{R}^d$ is of the form of a univariate function evaluated componentwise on the input, is the best neural network approximation for $f$ of a given size a parallel construction?

Another future direction is: To what extent do training methods for generative networks relate to these results? The results in this paper are only representational; they provide proof of what is possible with hand-chosen weights. One could experiment with training methods to see whether they create the "space-filling" property that is necessary for optimal increase of noise dimension. Alternatively, one could experiment with real-world datasets to see if changing the noise distributions while simultaneously growing or shrinking the network leaves the accuracy of the method unchanged. We ran some simple initial experiments measuring how well GANs of different architectures and noise distributions learned MNIST generation, and we found them inconclusive; in particular, we could not be certain if our empirical observations were a consequence purely of representation, or some combination of representation and training.

**Acknowledgements**

The authors are grateful for support from the NSF under grant IIS-1750051, and for a GPU grant from NVIDIA.

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
