[Supplementary Material]

# A Omitted proofs

## A.1 Proof of Theorem 2

Intuitively, our construction works as follows: Each output of the network will be a tentmap evaluated on one of the inputs. This will fill the output space in such a way that voxels within the unit cube of a certain size are all assigned equal mass.

Before we begin, we will define the tentmap formally, and make a few observations about it.

**Definition 2.** *The $k$-piece tentmap $t_k$ is the piecewise affine function from $[0,1]$ to $[0,1]$ defined as*

$$t_k(x) = \begin{cases} kx - \lfloor kx \rfloor & \text{for } \lfloor kx \rfloor \text{ even,} \\ 1 - kx + \lfloor kx \rfloor & \text{for } \lfloor kx \rfloor \text{ odd.} \end{cases}$$

We first note that $t_k$ can be implemented by a $k$-node, 2-layer network as

$$t_k(x) = \sigma(kx) + \sum_{i=1}^{k-1} 2(-1)^i \sigma(kx - i).$$

To see this, note that $\sigma$ is only nonzero when its argument is positive, and in this case it is equal to its input, so the sum can be rewritten (for $x \in [0,1]$) as

$$t_k(x) = kx + \sum_{i=1}^{\lfloor kx \rfloor} 2(-1)^i (kx - i).$$

For $\lfloor kx \rfloor$ even, we cancel out adjacent pairs of the sum to $-1$, leaving us with $kx + \frac{\lfloor kx \rfloor}{2}(-2) = kx - \lfloor kx \rfloor$. For $\lfloor kx \rfloor$ odd, we cancel out adjacent pairs leaving out the first term, leaving us with $kx - 2(kx - 1) - \frac{\lfloor kx \rfloor - 1}{2}(-2) = 1 - kx + \lfloor kx \rfloor$.

We also see the identity [see Telgarsky, 2016, Lemma 3.11]

$$t_j(t_k(x)) = t_{jk}(x)$$

since along each interval $[i/k, (i+1)/k]$, we get a copy of $t_j$ or its reflection.

The construction at the heart of Theorem 2 uses iterated tentmaps to transfer an $n$-dimensional distribution on the unit cube onto a space-filling curve on the unit cube in $d$ dimensions. To this end, we will prove a lemma justifying that this sort of space filling curve gives a bound in Wasserstein distance.

**Lemma 5.** *Let $f : [0,1]^n \to [0,1]^d$ be the map which takes $(x_1, x_2, \ldots, x_n)$ to the point*

$$\left( t_1(x_1), t_k(x_1), t_{k^2}(x_1), \ldots, t_{k^{d_1-1}}(x_1), t_1(x_2), \ldots, t_{k^{d_2-1}}(x_2), t_1(x_n), \ldots, t_{k^{d_n-1}}(x_n) \right),$$

*where $k, d_1, d_2, \ldots, d_n$ are positive integers with $\sum_{i=1}^n d_i = d$. Then*

$$W\left( f\#U([0,1]^n), U([0,1]^d) \right) \le \frac{\sqrt{d}}{k}.$$

*Proof.* Call a subinterval in $[0,1]$ a $k^i$-*interval* if it is of the form

$$I_a^i = (ak^{-i}, (a+1)k^{-i})$$

Where $a$ is an integer. We claim that the function $f$ maps each input box of the form

$$I_{a_1}^{d_1} \times \cdots \times I_{a_n}^{d_n}$$

to a subset of a distinct output box of the form

$$I_{b_1}^1 \times \cdots \times I_{b_n}^1.$$

To see this, we must show now that any two points in the same input box in $[0,1]^n$ map to the same output box in $[0,1]^d$, but that points from different boxes in $[0,1]^n$ map to different boxes

in $[0,1]^d$. For an input box parameterized by $(a_i)_{i=1}^n$, we note that $t_{k^e}$ maps a $k^{d_i}$-interval $I_{a_i}^{d_i}$ to a $k^{d_i-e}$-interval for all $0 \le e < d_i$ and to the interval $[0,1]$ for $e = d_i$. Thus, for two points $x, y \in [0,1]^n$, if $x_i, y_i$ fall in the same interval $I_{a_i}^{d_i}$, then for each coordinate $j$ corresponding to coordinate $i$, the $j$th coordinate of $f(x)$ and $f(y)$ will fall in the same interval $I_{b_j}^1$. If $x_i$ and $y_i$ fall in different intervals, then there will be a $j$ corresponding to $i$ such that the $j$th component of $f(x)$ and $f(y)$ fall in different $k$-intervals. Specifically, if $I_{a_i}^e$ contains both $x_i$ and $y_i$ but no $k^{e+1}$ interval contains both, then $f(x)_j$ and $f(y)_j$ will fall in different $k$-intervals where $j$ is the $e$th component of the output corresponding to $i$.

Since there are $k^d$ boxes of both the input and the output space, we see that there must be a 1-to-1 mapping from the boxes in $[0,1]^n$, to the boxes in $[0,1]^d$ containing their images under $f$. Since both types of boxes have measure $k^{-d}$ in the uniform measures on their respective unit cubes, there exists a coupling $\pi \in \Pi(f\#U([0,1]^n), U([0,1]^d))$ such that $\pi$ is supported on pairs $(x,y)$ belonging to the same box of the latter type. Then, since $|x - y| \le \frac{\sqrt{d}}{k}$ for any two points in a cube of side length $k^{-1}$, it suffices to choose any $\pi$ which arbitrarily associates points in the same cube, and the desired bound on Wasserstein distance follows.

$\square$

Now that we see how the tentmap-based function $f$ can achieve a low Wasserstein distance, the proof of Theorem 2 follows by writing $f$ as a ReLU network.

*Proof of Theorem 2.* Our network is designed as follows: Each layer of the network has two types of nodes: $d$ are "carry-forward nodes" which copy forward the value of a specific output once it is generated by a layer, and the remainder are "space-filling nodes", which compute high-frequency tent maps. We first set aside the $d$ nodes in each layer (a total of $dL$ nodes) to use as carry-forward nodes and use $x_{l,i,\texttt{carry}}$ to denote the carry-forward node in layer $l$ for output $i$. The remaining $N - dL$ space-filling nodes each correspond to a specific input component. We will call $n_{l,i}$ the number of space-filling nodes corresponding to input $i$ in layer $l$. We use $x_{l,i,j,\texttt{space}}$ to denote the input of the $j$th node corresponding to input $i$ in layer $l$.

We define the weights of the nodes such that the pre-ReLU activation of the $j$th node corresponding to input $i$ in layer $l$ is

$$x_{l,i,1,\texttt{space}} := n_{l,i} t_{k_{l,i}}(x_i) - (j-1),$$

where $k_{l,i} = \prod_{m=1}^{l-1} n_{m,i}$.

We see by induction on $l$ that it is possible to have the activations thus: For $l = 1$, we have $k_{l,i} = 1$, so the tent map we are replicating is the identity, and all the activations are affine functions of the input:

$$x_{1,i,j,\texttt{space}} = n_{l,i}(x_i) - (j-1).$$

For $l > 1$, we have (using the inductive hypothesis and the sum form of the tentmap) the tentmap $t_{n_{l,i}}$ as an affine combination of the post-ReLU activations of the previous layer evaluated on $t_{k_{l,i}}(x_i)$. Thus, using the product rule, we can obtain the value

$$t_{n_{l,i}}(t_{k_{l,i}}(x_i)) = t_{k_{l+1,i}}(x_i)$$

as an affine combination of layer $l$. And since this value can be computed in layer $l$, so can the affine transformations

$$n_{l+1,i} t_{k_{l+1,i}}(x_i) - (j-1) = x_{l,i,j,\texttt{space}}$$

for any value of $n_{l+1,i}$ and $j$.

Now that we have established how the space-filling nodes implement the tentmap function, we specify how these tentmaps feed in to the output. For input vector $(x_1, x_2, \ldots, x_n)$ the output vector will be of the form

$$\left( t_1(x_1), t_k(x_1), t_{k^2}(x_1), \ldots, t_{k^{\lceil d/n \rceil - 1}}(x_1), t_1(x_2), \ldots, t_1(x_n), \ldots, t_{k^{\lfloor d/n \rfloor - 1}}(x_n) \right),$$

where $k$ is an whole number depending on $N, L, n, d$. The $d$ outputs are split evenly among the $n$ inputs, with each output manifesting as a tentmap evaluated on its designated input. Each input has at most $\lceil \frac{d}{n} \rceil$ and at least $\lfloor \frac{d}{n} \rfloor$ output nodes taking the form of a tentmap of that input. Note that we

allow the trivial tentmap $t_1$, which is just the identity on $[0, 1]$, and our construction has each input $x_i$ with exactly one output of the form $t_1(x_i)$. Our goal is to make $k$ as large as possible with the limited number of carry-forward nodes and layers available, and then to prove that the Wasserstein accuracy of this construction decreases quickly with $k$.

We split the $N - dL$ space-filling nodes among the inputs in proportion with the number of outputs that input is responsible for. Thus, each output should get at least $\lfloor \frac{N-dL}{d} \rfloor$ nodes. Furthermore, each output that computes a nontrivial tentmap of its input is associated with a run of $\lfloor \frac{L}{\lceil \frac{d-n}{n} \rceil} \rfloor$ layers over which to distribute these nodes, so that the nodes for a certain input don't go over the total number of layers. With this in place, using the construction described above, we can guarantee a $k$ value of

$$ k = \left\lfloor \frac{\left\lfloor \frac{N-dL}{d} \right\rfloor}{\left\lfloor \frac{L}{\lceil \frac{d-n}{n} \rceil} \right\rfloor} \right\rfloor^{\left\lfloor \frac{L}{\lceil \frac{d-n}{n} \rceil} \right\rfloor}, $$

or to lower bound this with a more manageable expression,

$$ k \geq \left\lfloor \frac{\left\lfloor \frac{N-dL}{d} \right\rfloor}{\frac{L}{\lceil \frac{d-n}{n} \rceil}} \right\rfloor^{\lfloor \frac{nL}{d} \rfloor} \geq \left\lfloor \left( \frac{d-n}{n} \right) \frac{N - dL + d}{dL} \right\rfloor^{\lfloor \frac{nL}{d} \rfloor}. $$

Thus, we have the Wasserstein distance upper bound

$$ W(f \# U([0,1]^n), U([0,1]^d)) \leq \sqrt{d} \left\lfloor \left( \frac{d-n}{n} \right) \frac{N - dL + d}{dL} \right\rfloor^{-\lfloor \frac{nL}{d} \rfloor}. $$

$\square$

## A.2 Proof of Lemma 1

A concurrent proof of this theorem appears in Zhang et al. [2018], based on tropical geometry. Our proof is based on a lemma about how many different orthants an $n_0$-dimensional hyperplane can intersect, which turns out to be exponential in $n_0$. We then inductively track how many affine pieces exist in each layer of the network.

*Proof.* The proof requires a lemma:

**Lemma 6.** *A $k$-dimensional hyperplane $P$ in $\mathbb{R}^n$ intersects at most $\sum_{j=0}^{k} \binom{n}{j}$ orthants.*

This lemma follows from [Anthony and Bartlett, 2009, Lemma 3.3]. We assume without loss of generality that our plane is not parallel to any unit vector in $\mathbb{R}^n$. We then consider the $k + 1$ dimensional space containing $P$ and the origin as a copy of $\mathbb{R}^k$, and we project the $n$ unit vectors of the ambient $\mathbb{R}^n$ into this copy of $\mathbb{R}^{k+1}$. Applying [Anthony and Bartlett, 2009, Lemma 3.3], the perpendicular spaces of the vector projections split the copy of $\mathbb{R}^{k+1}$ into $2 \sum_{j=0}^{k} \binom{n}{j}$ connected components. These perpendicular spaces correspond to the separating planes of the orthants in $\mathbb{R}^n$, and since $P$ touches half of the connected components, it intersects $\sum_{j=0}^{k} \binom{n}{j}$ orthants in total.

With this lemma, we can now prove the theorem. We proceed by induction on $L$. In the $L = 0$ case, there are no nonlinearities in this network, and so $f$ is affine on its entire input.

In the inductive case, we consider $f$ computed by an $L + 1$ layer ReLU network. Let $g : \mathbb{R}^{n_0} \to \mathbb{R}^{n_L}$ represent the function computed by the first $L - 1$ hidden layers of $f$, outputting the last hidden layer of $f$. That is, $g = A_L \circ \sigma_{n_{L-1}} \circ \cdots \circ \sigma_{n_2} \circ A_2 \circ \sigma_{n_1} \circ A_1$. By the inductive hypothesis, $\mathbb{R}^{n_0}$ can be partitioned into

$$ N_A(g) \leq \prod_{i=1}^{L-1} \left( \sum_{j=0}^{n_0} \binom{n_i}{j} \right) $$

convex parts $S_1, \cdots S_{N_A(g)}$ such that $g$ is affine on each. For any of these convex regions $S_k$, the image of $g(S_k)$ is a convex set in $\mathbb{R}^{n_L}$. Consider the partition of $\mathbb{R}^{n_0}$ obtained by dividing each $S_k$ into subpieces according to the orthant of a points image under $g$. Because each orthant is convex, the preimage of each orthant under the restriction of $g$ to $S_k$ (which is affine) is also convex. Moreover, the function $f = A_{L+1} \circ \sigma_{n_L} \circ g$ is affine on each of these subpieces, because $g$ is affine on the subpieces and $\sigma_{n_L}$ is affine on each orthant (and $A_{L+1}$ is affine). Finally, since $g(S_k)$ is an affine image of a subset of $\mathbb{R}^{n_0}$, it lies in a $n_0$-dimensional hyperplane in $\mathbb{R}^{n_L}$, which can intersect at most $\sum_{j=0}^{n_0} \binom{n_L}{j}$ orthants. Thus, the subdivision step divides each $S_k$ into at most $\sum_{j=0}^{n_0} \binom{n_L}{j}$ subpieces. We therefore get

$$N_A(f) \le N_A(g) \sum_{j=0}^{n_0} \binom{n_L}{j} \le \prod_{i=1}^{L} \left( \sum_{j=0}^{n_0} \binom{n_i}{j} \right).$$

We also note the upper bound on the sum

$$\sum_{j=0}^{n_0} \binom{n_i}{j} \le \binom{n_i + n_0}{n_0} \le \frac{(n_i + n_0)^{n_0}}{n_0!} \le \left( e \frac{n_i + n_0}{n_0} \right)^{n_0}.$$

If we substitute this in above, we get the bound

$$N_A(f) \le \prod_{i=1}^{L} \left( e \frac{n_i + n_0}{n_0} \right)^{n_0}$$

and since (keeping the total number of nodes fixed) this product is maximized when all layers have the same number of nodes, we get

$$N_A(f) \le \prod_{i=1}^{L} \left( e \frac{N}{n_0 L} + e \right)^{n_0} = \left( e \frac{N}{n_0 L} + e \right)^{n_0 L}.$$

This proves the claim.

$\square$

## A.3 Proof of Theorem 3

The proof of this theorem comes in a few parts:

- We introduce a new notation to capture the idea of a Wasserstein distance between a distribution and a set.

- We prove a lemma about how distances between certain well-behaved distributions and hyperplanes can be lower bounded.

- We break the range of $f$ and the target distribution up into a collection of hyperplanes and distributions that can be handled by the lemma.

First, let us specify our idea of a Wasserstein distance between a distribution and a set:

**Definition 3.** *For a distribution $\mu$ and a closed, convex set $S$ on $\mathbb{R}^d$, we define the Wasserstein distance of the distribution from the set as*

$$W(\mu, S) := \inf_{\pi \in \Pi(\mu, S)} \int |x - y| d\pi(x, y) = \inf_{\nu \in \Pi_S} W(\mu, \nu)$$

*where $\Pi(\mu, S)$ is the set of joint distributions having $\mu$ as left marginal and right marginal supported on $S$, and where $\Pi_S$ is the set of all distributions supported on $S$.*

We can immediately note that an alternative way to view this definition is

$$W(\mu, S) = \int d(x, S) d\mu(x),$$

where $d(x, S) := \inf_{y \in S} |x - y|$ represents the distance of $x$ from $S$. To see this, we claim that there is an optimal coupling $\pi^* \in \Pi(\mu, S)$ which attains the minimum $\int d(x, y) \, d\pi^*(x, y) = W(U_B, S)$ and

Figure 2: Diagram of example $B$ (the ball in blue) and $R^*$ (the rectangle in red). Here $n = 1$ and $d = 2$.

which is supported on $(x, y)$ pairs where $y$ is the unique (since $S$ is closed and convex) closest point in $S$ to $x$. To see why such a coupling is optimal, note that for any other distribution $\pi \in \Pi(\mu, S)$,

$$\int |x - y| \, d\pi(x, y) \geq \int \inf_{y \in S} |x - y| \, d\pi(x, y) = \int |x - y| \, d\pi^*(x, y).$$

With this new definition, we move on to a lemma which lower bounds the Wasserstein distance between a uniform distribution on an arbitrary bounded measurable set and a hyperplane:

**Lemma 7.** *Let $B \subseteq B_0 \subseteq \mathbb{R}^d$ where $B_0$ is a ball of radius $l$, and $B$ has measure $m(B)$. Let $S$ be an $n$-dimensional hyperplane in $\mathbb{R}^d$. Then the Wasserstein distance between the uniform distribution on $B$ and the plane $S$ has the following lower bound:*

$$W(U_B, S) \geq \frac{d - n}{d - n + 1} \cdot \left( \frac{\Gamma(\frac{d-n}{2} + 1)\Gamma(\frac{n}{2} + 1)}{\pi^{\frac{d}{2}}} l^{-n} m(B) \right)^{1/(d-n)}$$

*Proof.* Using our new Wasserstein distance definition, we see that

$$W(U_B, S) = \frac{1}{m(B)} \int_B d(x, S) dx, \tag{3}$$

where $m(B)$ represents the Lebesgue measure of $B$.

We will now lower bound the integral over $B$ on the right hand side. We know $B$ is contained in $B_0$ of radius $l$ (centered at $x_0$, say), and the orthogonal projection $\text{Proj}_S(B)$ of $B$ onto the plane $S$ is, therefore contained in $R_1 := \text{Proj}_S(B_0)$, which is a ball of radius $l$ on the space $S$ (centered at $\text{Proj}_S(x_0)$). In the orthogonal complement space to $S$, define $R_2 \subset S^\perp$ as the ball which is centered on $S$ and has radius $r$ such that $m(R_1) \cdot m(R_2) = m(B)$ and define $R^*$ to be the Cartesian product of these balls in $d$-dimensional space, so that

$$m(R^*) = m(R_1 \times R_2) = m(R_1) \cdot m(R_2) = m(B). \tag{4}$$

We will see that this $R^*$ can replace $B$ in Equation 3 to provide the desired lower bound for the expression. From $m(B) = m(R^*)$, we get

$$m(R^* \setminus B) = m(R^* \cup B) - m(B) = m(R^* \cup B) - m(R^*) = m(B \setminus R^*).$$

Since $\text{Proj}_S(B) \subseteq R_1$, we have $B \subseteq R_1 \times S^\perp$, and since $R^*$ consists of all points in $R_1 \times S^\perp$ with distance $\leq r$ from $S$, for any $x \in B \setminus R^*$, we have $d(x, S) \geq r$. On the other hand, $d(x, S) \leq r$ for $x \in R^* \setminus B$ (since all elements of $R^*$ are within $r$ of $S$). Thus, we get a lower bound on the integral

from Equation 3

$$\int_B d(x,S)dx = \int_{B \cap R^*} d(x,S)dx + \int_{B \setminus R^*} d(x,S)dx$$

$$\geq \int_{B \cap R^*} d(x,S)dx + r \cdot m(B \setminus R^*)$$

$$= \int_{B \cap R^*} d(x,S)dx + r \cdot m(R^* \setminus B)$$

$$\geq \int_{B \cap R^*} d(x,S)dx + \int_{R^* \setminus B} d(x,S)dx$$

$$= \int_{R^*} d(x,S)dx,$$

and so

$$W(U_B, S) \geq \frac{1}{m(B)} \int_{R^*} d(x,S)dx. \tag{5}$$

(Note that we haven't given up much at this point; if our only restriction on $B$ was that its orthogonal projection was bounded by $l$, then we could have $R^* = B$ and the above inequality would be tight. As it is, if $l$ is much greater than $r$, then $B$ may share much overlap with $R^*$ anyway.)

We can decompose the above integral over $R^*$ in Equation 5 into the components parallel and perpendicular to $S$:

$$\int_{R^*} d(x,S)dx = m(R_1) \cdot \int_{R_2} d(x,S)dx. \tag{6}$$

The integral $\int_{R_2} d(x,S)dx$ is equivalent to the rotationally symmetric integral around the origin

$$\int_{R_2} d(x,S)dx = \int_{B_r} |x|dx, \tag{7}$$

where $B_r$ is the origin-centered ball of radius $r$ in $(d-n)$ dimensions. Intuitively, in a high dimensional space, most of the volume of a ball lies near its edge, so we can expect this integral to come out to about $r$ times the volume of the ball. We can evaluate the integral precisely by subtracting out $\int_{B_r} r - |x|dx$ and using the general formula that the volume of a cone (with $(d-n)$-dimensional base) is $\frac{1}{d-n+1}$ that of the cylinder with the same base and height:

$$\int_{B_r} |x|dx = \int_{B_r} rdx - \int_{B_r} r - |x|dx = r \cdot m(B_r) - r \cdot \frac{1}{d-n+1}m(B_r) = r \cdot m(B_r) \cdot \frac{d-n}{d-n+1}. \tag{8}$$

Putting together Equations (6) to (8), we get the integral from Equation 5 in terms of $r$, $m(B)$, $d$, and $n$:

$$\int_{R^*} d(x,S)dx = r \cdot m(R_1) \cdot m(R_2)\frac{d-n}{d-n+1} = r \cdot m(B) \cdot \frac{d-n}{d-n+1}.$$

All we need to complete the bound is to compute $r$. Recall we have $R_1$ as an $n$-dimensional ball of radius $l$ parallel to $S$, and $R_2$ is a $(d-n)$-dimensional ball of radius $r$ orthogonal to $S$ and centered on $S$. We use the fact that $m(R_1) \cdot m(R_2) = m(B)$ to get

$$\left( \frac{\pi^{\frac{d-n}{2}}}{\Gamma(\frac{d-n}{2}+1)}r^{d-n} \right) \left( \frac{\pi^{\frac{n}{2}}}{\Gamma(\frac{n}{2}+1)}l^n \right) = m(B),$$

which after solving for $r$ gives

$$r = \left( \frac{\Gamma(\frac{d-n}{2}+1)\Gamma(\frac{n}{2}+1)}{\pi^{\frac{d}{2}}}l^{-n}m(B) \right)^{1/(d-n)}.$$

Substituting this value for $r$ in Equation 5 gives

$$W(U_B, S) \geq \frac{1}{m(B)} \int_{R^*} d(x, S) dx$$

$$\geq \frac{d-n}{d-n+1} \cdot \left( \frac{\Gamma(\frac{d-n}{2} + 1)\Gamma(\frac{n}{2} + 1)}{\pi^{\frac{d}{2}}} l^{-n} m(B) \right)^{1/(d-n)},$$

as desired. $\qquad\square$

Now that we have this lemma regarding distance to hyperplanes, we prove the theorem by applying the lemma to the planes on which the range of our piecewise affine function lies.

*Proof of Theorem 3.* We first note that since $P$ can be any distribution on $\mathbb{R}^n$, $f\#P$ can be any distribution on the range of $f$. Therefore, it suffices to lower bound the Wasserstein distance between the distribution $U_B$ and the range of $f$.

Since $f$ is piecewise affine, its range is a subset of the union of $N_A(f)$ $n$-dimensional hyperplanes in $\mathbb{R}^d$, which we name $S_1, \cdots, S_{N_A(f)}$. We call the union of these planes $S$, and we note that

$$W(U_B, \mathrm{Range}\, f) \geq W(U_B, S),$$

since any distribution supported on $\mathrm{Range}\, f$ is supported on its superset $S$.

The Wasserstein distance of $U_B$ to this set $S$ is lower bounded by the integral of $d(x, S)$ over $U_B$, since this lower bounds the integral for any $\pi \in \Pi(U_B, S)$. In fact, this is an equality, since the following correspondence gives us a $\pi$ that achieves this minimum: For each $i$, let $B_i \subseteq B$ consist of all $y$ that are nearer to $S_i$ than any other $S_j$ choosing the smaller index in case of ties. That is,

$$B_i = \{x \in B : i = \arg\min_j d(x, S_j)\}.$$

This makes each $B_i$ a measurable set such that for $x \in B_i$, we have

$$\inf_{y \in S} |x - y| = d(x, S_i),$$

and by choosing $\pi \in \Pi(U_B, S)$ supported on $(x, y)$ pairs where $y$ is the closest point in $S$ to $x$, (choosing the minimum index when ambiguity arises), we get

$$W(U_B, S) = \inf_{\pi \in \Pi(U_B, S)} \int |x - y| d\pi(x, y)$$

$$= \int d(x, S) dU_B(x)$$

$$= \frac{1}{m(B)} \sum_i \int_{B_i} d(x, S_i) dy.$$

As we noted, these integrals can be expressed in terms of the Wasserstein distances of the uniform distributions on the $B_i$ to their respective $S_i$:

$$W(U_B, S) = \frac{1}{m(B)} \sum_i m(B_i) \cdot W(U_{B_i}, S_i).$$

We apply the lemma to lower bound this integral for each $i$, whereby

$$W(U_B, S) \geq \frac{1}{m(B)} \sum_i m(B_i) \cdot \frac{d-n}{d-n+1} \cdot \left( \frac{\Gamma(\frac{d-n}{2} + 1)\Gamma(\frac{n}{2} + 1)}{\pi^{\frac{d}{2}}} l^{-n} m(B_i) \right)^{1/(d-n)}.$$

Since the summand is convex in $m(B_i)$, Jensen's inequality inequality allows replacing $m(B_i)$ with $m(B)/N_A$, thus

$$\geq \frac{1}{m(B)} \sum_i \frac{m(B)}{N_A} \cdot \frac{d-n}{d-n+1} \cdot \left( \frac{\Gamma(\frac{d-n}{2}+1)\Gamma(\frac{n}{2}+1)}{\pi^{\frac{d}{2}}} l^{-n} \frac{m(B)}{N_A} \right)^{1/(d-n)}$$

$$= N_A \frac{1}{N_A} \cdot \frac{d-n}{d-n+1} \cdot \left( \frac{\Gamma(\frac{d-n}{2}+1)\Gamma(\frac{n}{2}+1)}{\pi^{\frac{d}{2}}} l^{-n} \frac{m(B)}{N_A} \right)^{1/(d-n)}$$

$$= \frac{d-n}{d-n+1} \cdot \left( \frac{\Gamma(\frac{d-n}{2}+1)\Gamma(\frac{n}{2}+1)}{\pi^{\frac{d}{2}}} l^{-n} \frac{m(B)}{N_A} \right)^{1/(d-n)},$$

which is of the desired form. □

### A.4 Proof of Theorem 5

*Proof.* We will approximate $x^k$ inductively by multiplying the approximation for $x^{k-1}$ with $x$. We will ensure that we approximate each $x^k$ to within $\frac{\epsilon}{(2M)^{n-k}}$ (assuming $M$ is at least 1). If we approximate the multiplication by $x$ function to within $\frac{\epsilon}{2(2M)^{n-k}}$, and consider that multiplication by $x$ is $M$-Lipschitz, then using Lemma 8, we have that if $x^{k-1}$ is approximated to within $\frac{\epsilon}{(2M)^{n-k+1}}$ then $x^k$ will be approximated to within

$$M\frac{\epsilon}{(2M)^{n-k+1}} + \frac{\epsilon}{2(2M)^{n-k}} = \frac{\epsilon}{(2M)^{n-k}}.$$

By induction, this construction will indeed approximate all $x^k$ to within our specified accuracy. Analyzing the size of this network, we see that the network module computing $x^{k-1} \cdot x = x^k$ will require

$$O(\ln(2(2M)^{n-k}/\epsilon) + \ln(M^k)) = O((n-k)\ln(2M) + (n-k) + \ln(1/\epsilon) + k\ln(M))$$

nodes. Summing this over $k = 1$ to $n-1$ produces $\mathrm{poly}(n, \ln(M), \ln(1/\epsilon))$. □

### A.5 Proof of Theorem 6

*Proof.* As mentioned before, $\Phi$ has the series representation

$$\Phi(z) = \frac{1}{2} + \frac{1}{\sqrt{2\pi}} \sum_{k=0}^{\infty} \frac{(-1)^k z^{2k+1}}{k!(2k+1)2^k}.$$

We consider the truncated sum

$$\Phi_n(z) = \frac{1}{2} + \frac{1}{\sqrt{2\pi}} \sum_{k=0}^{n} \frac{(-1)^k z^{2k+1}}{k!(2k+1)2^k},$$

where we set $n = \max\{2eM^2 - 1, \log_2(2/\epsilon)\}$. This guarantees that for $x \in [-M, M]$, the error incurred by truncating the sum is

$$|\Phi(z) - \Phi_n(z)| = \frac{1}{\sqrt{2\pi}} \left| \sum_{k=n+1}^{\infty} \frac{(-1)^k z^{2k+1}}{k!(2k+1)2^k} \right|$$

$$\leq \sum_{k=n+1}^{\infty} \frac{|M|^{2k+1}}{k!}. \qquad \because \text{Stirling's inequality}$$

$$\leq \sum_{k=n+1}^{\infty} \frac{M^{2k+1}}{\sqrt{2\pi}k^{k+\frac{1}{2}}e^{-k}}$$

$$= \sum_{k=n+1}^{\infty} \frac{1}{\sqrt{2\pi e}} \left( \frac{eM^2}{k} \right)^{k+\frac{1}{2}}.$$

Since we chose $n + 1 \geq 2eM^2$, we have

$$\leq \sum_{k=n+1}^{\infty} \left(\frac{1}{2}\right)^k$$
$$= 2^{-n},$$

and since we chose $n \geq \log_2(2/\epsilon)$,

$$\leq \frac{\epsilon}{2}.$$

So the total error we get by omitting these terms is no more than $\frac{\epsilon}{2}$. We approximate each $z^{2k+1}$ to within $\epsilon/(2n + 2)$, and then multiply each $x^i$ by its Maclaurin coefficient. Since each coefficient is no more than 1 in absolute value, the errors in each of the Maclaurin terms is no more than $\epsilon/(2n + 2)$. We can therefore add all $n+1$ of these Maclaurin terms and get an error less than $\frac{\epsilon}{2}$ from the truncated series $\tilde{f}$, and a total error no more than $\epsilon$ from the function $\Phi$ in the interval $[-M, M]$. Applying Theorem 5, approximating the $x^i$ to this accuracy requires

$$\mathrm{poly}(2n + 1, \ln(M), \ln(1/\epsilon)) = \mathrm{poly}(M, \ln(1/\epsilon))$$

nodes. Take $M$ sufficiently large that $1 - \Phi(M/2) < \epsilon$, which can be done with $M = O(\log(1/\epsilon))$. Then, add a component to the neural network that interpolates between this approximation on $[-M, M]$ and 1 for $z > M/2$ and 0 for $x < -M/2$. This guarantees the network is accurate for all values of $z$. Note that we can guarantee the range of this approximation falls in $[0, 1]$, by adding a gadget that clamps the output to this interval. $\qquad\square$

### A.6 Proof of Lemma 3

*Proof.* We construct a ReLU/Step network which contains $t$ copies of the neural network approximating $f$, as well $3t + 3$ nodes called $x_{i,\mathtt{low}}, x_{i,\mathtt{mid}}, x_{i,\mathtt{high}}$ for $i$ in $\{0, \ldots, t\}$. The network assigns the initial values

$$x_{0,\mathtt{low}} := a, \qquad x_{0,\mathtt{high}} := a,$$

and for all values of $i$, we compute

$$x_{i,\mathtt{mid}} := \frac{x_{i,\mathtt{low}} + x_{i,\mathtt{low}}}{2}.$$

Let $y \in [c, d]$ be the input to our network for computing $f^{-1}$. For $0 \leq i < t$, we let $x_{i,\mathtt{mid}}$ be the input to the $i$th copy of the network computing $f$, and call the output node of this copy $y_{i,\mathtt{mid}}$, and we correspondingly call. If $y_i \geq y$ (which we test using a step function activation), set

$$x_{i+1,\mathtt{low}} := x_{i,\mathtt{low}}, \qquad x_{i+1,\mathtt{high}} := x_{i,\mathtt{mid}},$$

and otherwise, set

$$x_{i+1,\mathtt{low}} := x_{i,\mathtt{mid}}, \qquad x_{i+1,\mathtt{high}} := x_{i,\mathtt{high}}.$$

We set the network output to be $x_{t,\mathtt{mid}}$.

By induction on the construction of these values, if $\tilde{f}$ is the approximation of $f$ given by the provided network, then the interval $[\tilde{f}(x_{i,\mathtt{low}}), \tilde{f}(x_{i,\mathtt{high}})]$ contains $y$ for each $i$. This implies that $x = f^{-1}(y)$ is in the interval $[f^{-1}(\tilde{f}(x_{t,\mathtt{low}})), f^{-1}(\tilde{f}(x_{t,\mathtt{high}}))]$, since $f$ is increasing. Moreover, since $f^{-1}$ is $L$-Lipschitz and $\tilde{f}$ is accurate to within $\epsilon$, the endpoints of this interval are within $\epsilon L$ of $x_{t,\mathtt{low}}$ and $x_{t,\mathtt{high}}$, so we know that the above interval is contained in

$$[x_{t,\mathtt{low}} - \epsilon L, x_{t,\mathtt{high}} + \epsilon L].$$

So $x$ is contained in this interval, but $x_{t,\mathtt{mid}}$ is the midpoint of this interval, so the maximum possible distance between $x$ and $x_{t,\mathtt{mid}}$ is half the length of the interval, which is $(b - a)2^{t+1} + \epsilon L$. $\qquad\square$

## A.7 Proof of Theorem 9

*Proof.* Applying Theorem 8 with $\epsilon = \epsilon_1$ (to be specified later), we approximate the inverse CDF of the normal distribution on $[\Phi(-\ln(1/\epsilon_1^2)), \Phi(\ln(1/\epsilon_1^2))]$. On the intervals $[0, \Phi(-\ln(1/\epsilon_1^2))]$ and $[\Phi(\ln(1/\epsilon_1^2)), 1]$, we set the output of the network to $-\ln(1/\epsilon_1^2)$ and $\ln(1/\epsilon_1^2)$ respectively (using step function activations to test if the input is in this range). Finally, we append a gadget computing the function $x \mapsto \max(-\ln(1/\epsilon_1^2), \min(x, \ln(1/\epsilon_1^2)))$, so that the output of our network $f$ lies in the range $[-\ln(1/\epsilon_1^2), \ln(1/\epsilon_1^2)]$. We now look to lower bound the Wasserstein for this generative network $f$.

$$W(f\#U([0,1]), \mathcal{N}) = \inf_{\pi \in \Pi} |x - y| d\pi(x, y).$$

We consider a coupling between $f\#U([0,1])$ and $\mathcal{N}$ with pairs of the form $(f(x), \Phi^{-1}(x))$, where $x \sim U([0,1])$:

$$\leq \int_0^1 |f(x) - \Phi^{-1}(x)| dx.$$

We now split this integral into three parts

$$= \int_A |f(x) - \Phi^{-1}(x)| dx + \int_{I \setminus A} |f(x) - \Phi^{-1}(x)| dx + \int_{[0,1] \setminus (I \cup A)} |f(x) - \Phi^{-1}(x)| dx$$

$$\leq m(A) \cdot 2\ln(1/\epsilon_1^2) + 1 \cdot \epsilon_1 + 2 \int_{\Phi(\ln(1/\epsilon_1^2))}^1 |f(x) - \Phi^{-1}(x)| dx.$$

Rewriting the integral on the tails,

$$= m(A) \cdot 2\ln(1/\epsilon_1^2) + 1 \cdot \epsilon_1 + 2 \int_{\ln(1/\epsilon_1^2)}^\infty 1 - \Phi(x) dx,$$

and since the normal CDF has exponentially small tails

$$= m(A) \cdot 2\ln(1/\epsilon_1^2) + O(\epsilon_1).$$

Now, choosing $m(A)$ sufficiently small,

$$= O(\epsilon_1),$$

and we can choose $\epsilon_1$ sufficiently small so that this is under $\epsilon$. Since $\epsilon_1$ is linear in $\epsilon$, the construction still uses $\mathrm{polylog}(1/\epsilon)$ nodes. □

## A.8 Proof of Proposition 2

*Proof.* Let $[0 = a_0, a_1], [a_1, a_2], \ldots, [a_{N_A - 1}, a_{N_A} = 1]$ be the intervals on which $f|_{[0,1]}$ is affine. The distribution given by $f\#U([0,1))$ is a mixture of $N_A$ distributions

$$f\#U([0,1]) = \sum_{i=0}^{N_A - 1} \frac{1}{a_{i+1} - a_i} f\#U([a_i, a_{i+1}]),$$

where the $f\#U([a_i, a_{i+1}])$ is either a uniform distribution on the interval $[f(a_i), f(a_{i+1})]$ (or $[f(a_{i+1}), f(a_i)]$), or if $f(a_i) = f(a_{i+1})$, it is a point distribution on $f(a_i)$. Since these distributions have CDFs which are nonlinear only at $f(a_i)$ values, the CDF of $f\#U([0,1])$ (which is the weighted sum of CDFs of these simple distributions), is piecewise affine with nonlinearities at $f(a_i)$. In other words, it is piecewise affine in $N_A + 2$ pieces. □

### A.9 Proof of Lemma 4

*Proof.* Define $g$ to be the function

$$g(x) = \max(0, \min(1, f(x))).$$

Since $f$ has $N_A$ affine pieces, and each of these can yield at most 3 pieces in $g$, $g$ has $3N_A$ affine pieces at most. By Safran and Shamir [2016, Theorem 7], since $\Phi$ has second derivative bounded away from 0 on an interval, we get

$$\int_a^b |\Phi(x) - g(x)|^2 dx \geq \frac{k}{(3N_A)^4} = \frac{K}{N_A^4}$$

for some constant $K$. Since $|\Phi(x) - g(x)| \leq 1$, we have

$$\int_a^b |\Phi(x) - f(x)| dx \geq \int_a^b |\Phi(x) - g(x)| dx \geq \frac{K}{N_A^4}.$$

$\square$

### A.10 Proof of Theorem 12

*Proof.* Each of the uniform variables has mean $1/2$ and variance $1/12$ so subtract $n/2$ and multiply the sum by $1/\sqrt{12}$ to normalize. The nonuniform version of the Berry-Esseen theorem [Pinelis, 2013] tells us that there is a constant $C$ such that the difference in CDF between this and the normal CDF at $t$ is no more than $\frac{C}{\sqrt{n}(1+|t|^3)}$. Since the integral of $1/(1+|t|^3)$ converges, the integral of this difference over all $\mathbb{R}$ is bounded by $O(\frac{1}{\sqrt{n}})$ and by Proposition 1, this gives the Wasserstein distance bound. $\square$

## B Additional Lemma

**Lemma 8.** *If we have $A, B, C$ subsets of Euclidean spaces and functions $f, \tilde{f} : A \to B$ and $g, \tilde{g} : B \to C$ such that*

- *For all $x \in A$, $|\tilde{f}(x) - f(x)| < \frac{\epsilon}{2L_g}$ (where $L_g$ is a Lipschitz constant of $g$)*

- *For all $y \in B$, $|\tilde{g}(y) - g(y)| < \frac{\epsilon}{2}$*

*then $|(\tilde{f} \circ \tilde{g})(x) - (f \circ g)(x)| \leq \epsilon$ for all $x \in A$.*

*Proof.* If $|\tilde{f}(x) - f(x)| < \epsilon_1 = \frac{\epsilon}{2L_g}$ and $|\tilde{g} - g| < \epsilon_2 = \frac{\epsilon}{2}$, then applying the triangle inequality gives us

$$\begin{aligned}
|(\tilde{g} \circ \tilde{f})(x) - (g \circ f)(x)| &= |(\tilde{g} \circ \tilde{f})(x) - (g \circ \tilde{f})(x) + (g \circ \tilde{f})(x) - (g \circ f)(x)| \\
&\leq |(\tilde{g} \circ \tilde{f})(x) - (g \circ \tilde{f})(x)| + |(g \circ \tilde{f})(x) - (g \circ f)(x)| \\
&\leq \epsilon_2 + L_g \epsilon_1 \\
&= \epsilon.
\end{aligned}$$

$\square$