[Reviews · NeurIPS 2018]

Reviewer 1



The paper presents a theoretical analysis on the presentational capabilities of generative network, which is limited to a full-connected network architecture with ReLU activation function. It gives detail discussion on different cases where the input dimension is less than, equal to or greater than output dimension. This paper is out of my area. Though I can follow most of the technical parts, I'm not familiar with the details in math and the assumptions. Overall it's an interesting reading to me, and the result can possibly shed light on generative networks training analysis. Clarity Overall I think the paper is organized. One suggestion on related works: mention (Box et al., 1958) in section 1.1 instead of in conclusion section. Minor: line 64 where citing “[Goodfellow et al., 2014]”, “Goodfellow et al. [2014]” to be consistent with other references format.

Reviewer 2



This paper examines the approximation of uniform and normal distributions using neural networks provided with a uniform or normal distribution at its input. The paper's main contributions are 1. the upper- and lower-bound for approximating higher dimensional uniform distributions with a lower dimensional one. The main idea is the use of an affine space-filling curve, which has been examined in a similar context before. 2. providing a neural network transformation to go from a uniform distribution to a normal and vice-versa. Overall, the paper is a pleasure to read and tackles the important question of the required size/depth required for a neural network to approximate a given distribution. Some minor notes: Section 2.2 is not an easy read without delving into the appendix. Perhaps some intuition can be provided for the statement in line 138. Lines 145-148 are also a bit terse. An added explanation of how the lower and upper-bound compare (as is done in line 33 of the intro) would make the strength of this section more apparent. Section 4 is somewhat trivial as it is stated. I would suggest adding a few sentences making the comparison with the result of Section 3.2 more explicit, so that the added benefit of more dimensions of uniform noise to approx. a normal becomes clear. If more space is required, perhaps consider reducing the size of Figure 1 or placing proof of Th. 11 in the appendix. Some typos: l 53 This idea [h]as l 104 How easy [is it] to l 190 good behavior <- vague l 269 we [...] get a bound

Reviewer 3



This paper is a joy to read. It addresses question whether GANs can map between simpler standard distributions and how well, and the role of width and depth of the neural net in the approximation. The paper is very well organized. Deep down, the argument boils down to that the ReLU neural network can divide the input space into small convex open subsets that are locally uniform and then map them (via affine transformations) to another small locally uniform set of the output. The measure remains the same after the mapping. The dificulty is: how small should these sets be? And how well can a given neural network do this task? I think the authors did a great job explaining the task. It was disapointing that there was no empirical results trying to validate the bounds. I have a personal bias against papers with theoretical bounds that show no empirical evidence of their results. Specifically in this case, where the examples are simple and concrete. Please also cite Egoroff’s theorem [Folland, 2013, Theorem 2.33] as Egorov's theorem [Kolmogorov and Fomin, 1975, pp. 290, Theorem 12]. The different spellings of the name tend to confuse readers familiar only with Kolmogorov's book. A. N. Kolmogorov and S. V. Fomin. Introductory Real Analysis. Dover, 1975. The statement that "It stands to reason that larger neural networks or networks with more noise given as input can represent more". As R (real set) has the same cardinality as R^n, n finite, the statement seems to be connected to a notion of smoothness in the measure. It reads a little odd and a small discussion on the reason why the input needs to be high-dimensional noise could help. --------- Read the rebuttal and the other two reviews. This is a good contribution.